

# The effect of different sowing dates on dry matter and nitrogen dynamics for winter wheat: an experimental simulation study

Kaizhen Liu[1], Chengxiang Zhang[1], Beibei Guan[1], Rui Yang[1], Ke Liu[1], Zhuangzhi Wang[1], Xiu Li[1], Keyin Xue[1], Lijun Yin[1] and Xiaoyan Wang[1,2]

[1] Agronomy College of Yangtze University, Yangtze University, Jingzhou, Hubei, China
[2] Engineering Research Center of Ecology and Agricultural Use of Wetland, Ministry of Education, Yangtze, Jingzhou, Hubei, China

## ABSTRACT

**Background**. Timely sowing is an important agronomic measure to ensure the normal germination, stable seedling establishment, and yield formation for winter wheat (*Triticum aestivum L.*). Delayed sowing frequently occurs in the current multi-cropping system and mechanized production of this crop. However, the ways in which different sowing dates affect yield and its potential mechanism is still unknown in the middle-lower Yangtze River Basin. We sought to provide a theoretical basis for these mechanisms to improve regional wheat production.

**Methods**. We investigated the wheat's yield differences in a two-year field study under different sowing dates and took into account related growth characteristics including meteorological conditions, growth period, tillers, dry matter accumulation (DMA), and nitrogen accumulation (NA). We used the logistic curve model to simulate DMA and NA dynamics of single stem wheat under different sowing dates. We then analyzed and compared wheat accumulation for different sowing dates.

**Results**. Our results showed that grain yield declined by $0.97 \pm 0.22\%$ with each one-day change (either early or delayed) in sowing beyond the normal sowing date. The yield loss could be explained by the inhibition of crop growth, yield components, biomass and nitrogen (N) production. The negative effects of delayed sowing were caused by environmental limitations including adverse weather factors such as low temperature during vegetative growth, shortened duration of various phases of crop development, and increased temperature during the grain-filling period. The grain yield gap decreased between the late and normal sowing periods owing to a compensatory effect between the highest average rates ($V_t$) and the rapid accumulation period ($T$) of DMA and NA for single stem wheat. The grain yield was maintained at 6,000 kg ha$^{-1}$ or more when the ratio of DMA at the mature-to-jointing stage ($M_D/J_D$) and the ratio of NA at the mature-to-jointing stage ($M_N/J_N$) was 4.06 ($P < 0.01$) and 2.49 ($P < 0.05$), respectively. The compensatory effect did not prevent the impact caused by delayed sowing, which caused biomass and N production to decrease. Physiological development reached a maximal accumulation rate ($T_m$) of NA earlier than DMA.

Corresponding authors
Lijun Yin, yinlijun326@163.com
Xiaoyan Wang,
wamail_wang@163.com

# INTRODUCTION

Wheat is one of the most widely-cultivated crops worldwide and wheat growers in China are the largest producers in the world, producing 17% of the world's total wheat (*Food and Agricultural of the United Nation, 2019*). Winter wheat accounts for approximately 95% of the total (winter and spring) wheat production in China (*Lu & Fan, 2013*; *Wu et al., 2014*; *Geng et al., 2019*). The middle-lower Yangtze River Basin is one of the main winter wheat growing regions in China due to its abundance of photothermal resources necessary for the rice-wheat rotation system (*Zhang et al., 2013*). The practice of rice-wheat crop rotation dominates in this area (*Liu et al., 2016*) and its grain output is vital to ensuring food security in China.

The sowing date is one of the most important factors affecting grain production and quality (*Ferrise et al., 2010*). The optimum sowing date depends on rainfall and temperature (*Jackson et al., 2000*) to maintain high grain yields. The rice-wheat rotation system is one of the world's most predominant agricultural production systems (*Gupta et al., 2003*). However, there are limitations to this planting pattern. In recent years, simple cultivation methods including mechanical transplanting and direct seeding of rice have shortened the seedling period. There has been a significant yield benefit to the delayed sowing of rice. Wheat crops are planted after rice crops and the late sowing of rice impacts the sowing date for wheat as well (*Xu et al., 2013*).

Global warming has significantly impacted agricultural production and has been the focus of many researchers (*Sun et al., 2015*; *Ding et al., 2015*). Simulation studies and observed data have shown that a significant decrease in the grown duration of winter wheat may be accompanied by a significant reduction in grain yield due to the evident warming trend (*He et al., 2014*; *He et al., 2015*; *Xiao et al., 2015*). Global warming over recent decades has created extended growing periods prior to wheat wintering, encouraging farmers to delay the sowing date for winter wheat (*Xiao et al., 2013*; *Xiao et al., 2015*). Studies have shown that this delay may increase, maintain, or decrease the grain yield of winter wheat (*Jalota et al., 2013*; *Ding et al., 2015*; *Yin, Dai & He, 2018*).

Previous research has indirectly suggested that sowing wheat late leads to poor crop conditions, even with optimal weather conditions (*Tester & Langridge, 2010*). Delayed sowing exposes crops to adverse conditions such as low temperatures during vegetative growth, resulting in a low germination rate, poor tillering ability, and low plant population (*Borràsgelonch et al., 2012*; *Fernanda et al., 2013*). Late sowing delays flowering and exposes crops to high temperatures during the grain-filling stage, thus accelerating reproductive development and reducing grain-filling (*Bailey-Serres et al., 2019*; *Dubey et al., 2019*). Late sowing also reduces dry matter and N accumulation in wheat crops (*Ehdaie & Waines, 2001*). Therefore, a delay in sowing wheat often has a negative impact on the seed germination process, tiller development, overall crop growth, and final yield (*Hussain et*

*al., 2017*; *Kaur, 2017*). It is important to note that a delay sowing during the optimal period does not have a serious negative impact on yield, as it may improve the assimilate allocation and N utilization efficiency for winter wheat (*Yin, Dai & He, 2018*; *Yin et al., 2019*).

The formation of crop yield is determined by the accumulation and distribution of dry matter (*Zheng, Xu & Wu, 2013*). Natural and human factors such as climate, soil, and field management practices all affect the dry matter accumulation (DMA) process and ultimately lead to yield differences. Nitrogen accumulation (NA) is the main nutrient affecting the grain yield and protein concentrations (*Ehdaie & Waines, 2001*). DMA and NA can be described using two models: the mechanism model and the empirical model (*Whisler et al., 1986*). In the empirical model, the logistic growth equation (*Wang et al., 2014*; *Jain, Agrawal & Singh, 2010*; *Royo & Blanco, 1999*) has certain biological significance and is widely used.

The logistic equation has been used to describe the accumulation of dry matter for winter wheat and summer maize as well as the cotton NA process (*Zhao et al., 2013*; *Xiao et al., 2014*; *Du et al., 2016*). The accumulation of dry matter and N is a continuous process that changes over time and is closely related to yield. There are significant differences seen over time and research sites. There have been few studies on the effects of the sowing date on the growth and yield of wheat in the Jianghan Plain of the middle-lower Yangtze River Basin in terms of DMA, NA, and yield. The objectives of this study were to (i) quantify the effect of different sowing dates on grain yield, yield components, tillers, and other agronomic traits of winter wheat, and (ii) utilize the logistic model to fit the DMA and NA process of winter wheat in different sowing dates. We quantitatively analyzed the growth process of wheat according to the derived quantity.

## MATERIAL AND METHODS

### Experimental site

Field experiments were conducted at the experimental farm station of Yangtze University (303°6′N, 1120°8′E), Jinzhou City, Hubei Province, China, during two growing seasons in 2018/2019 and 2019/2020. The farm station is located in the Jianghan Plain, which is characterized as a typical subtropical monsoon climate zone in the middle-lower Yangtze River Basin of China. Two-year field experiments were performed in the nearby fields. The field chosen for this study was previously managed as a summer-rice/winter-wheat double-cropping system. The daily average temperature and precipitation during the two-year growing seasons are shown in Fig. 1. Soil samples were collected at the start of the experiments. The soil was classified as sandy loam, and the main physicochemical properties before sowing wheat in 2018 and 2019 are shown in Table 1.

### Experimental design and crop management

A widely-planted winter wheat cultivar, Zhengmai 9023, was used in our field experiments. Seeds were sown by broadcasting at a rate of $15\,\mathrm{g\,m^{-2}}$ in 2018 and 2019 on 28 October (early sowing), 5 November (normal sowing), 13 November (late sowing), and 21 November (latest sowing) using a manual ditching drill with 25-cm row spacing. The plots were arranged in a completely random design with three replicates. Each plot included 25 rows

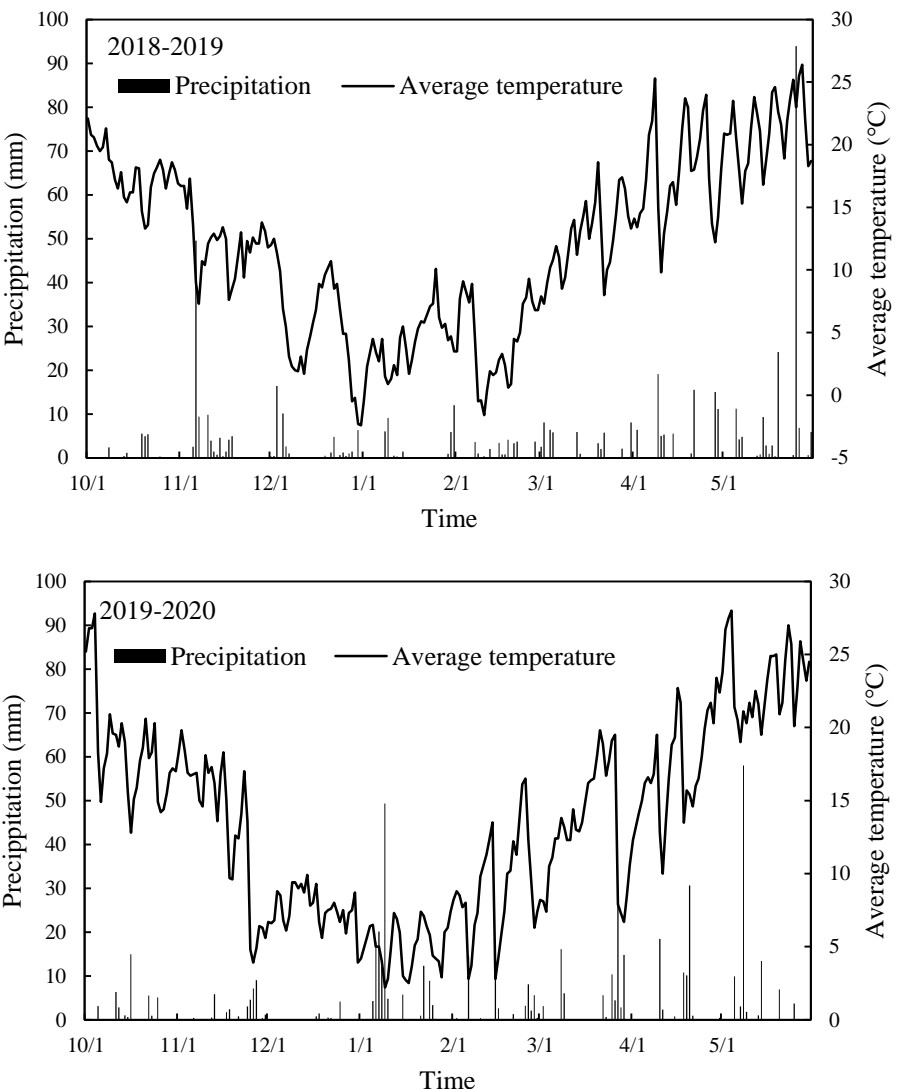

**Figure 1 Average temperature and precipitation over two growing seasons.** The top panel shows data from the 2018–2019 growing season, while the bottom panel shows data from the 2019–2020 growing season. Black lines indicate average temperatures in 2018.10–2019.5 and 2019.10–2020.5, respectively. Black bars indicate precipitation events during 2018.10–2019.5 and 2019.10–2020.5, respectively. Date was collected by the local meteorological bureau of Jingzhou.

**Table 1 The soil fertility status of the tested soil (0–20 cm).**

| Season | pH | Organic matter (g kg⁻¹) | Alkali hydrolyzed nitrogen (mg kg⁻¹) | Available phosphorous (mg kg⁻¹) | Available potassium (mg kg⁻¹) |
|---|---|---|---|---|---|
| 2018–2019 | 8.23 | 17.9 | 60.5 | 21.9 | 116.6 |
| 2019–2020 | 8.19 | 14.8 | 42.8 | 7.3 | 75.7 |

with 25-cm in-row spacing (2 m wide), 6 m in length. Basal fertilization of each subplot included N as urea, phosphorus as calcium superphosphate, and potassium as potassium chloride at rates of 90 kg ha$^{-1}$ N, 105 kg ha$^{-1}$P$_2$O$_5$, and 105 kg ha$^{-1}$ K$_2$O, respectively. An additional 90 kg ha$^{-1}$ of N was applied at the beginning of the joint growing stage (*Sun et al., 2019*; *Yin et al., 2020*). Fields were managed following the local cultural practices. Pests, weeds, and diseases were controlled chemically.

## Measurement items and methods

Two 0.5 m lines with uniform emergence were made at the three-leaf stage. These were randomly selected for fixed points in each plot. Tillers were counted before wintering and at the jointing, booting, flowering, and maturity stages in each plot with three repeats, respectively (*Zhou et al., 2020*).

Sampling for whole plant dry matter was conducted before wintering and at the jointing, booting, flowering, and maturity stages in each plot with three repeats at ground level. The samples were dried at 105 °C for 30 min and then at 80 °C in a fan-forced oven to a constant weight to determine the biomass (*Shah et al., 2020*). A sub-sample of 30 fertile shoots was separated into flag leaves, other leaves, stem sheaths, and ear tissues (glumes and grains at maturity). All separated plant material was oven-dried as described above, weighed, milled, and analyzed for total N concentration (Kjeldahl method, KDY-9820 Auto Distillation Unit, Beijing, China). NA was calculated by multiplying the N concentration (%) by dry weight. The NA of the single stem was calculated as the sum of N absorption of the single stem in different measured organs at each growing stage. This process was repeated three times (*Yin, Dai & He, 2018*).

Plants used to measure yield were harvested from a sampling area 2.0 m ×1.0 m (row length × row width) in each plot, using the method described by *Yin, Dai & He (2018)*. The grain was air-dried, weighed, and adjusted to standard 12% moisture content. This was considered to be the grain dry matter yield. Effective panicle per unit area was measured from a sampling area 1.0 m ×1.0 m (row length × row width). Thirty panicles were taken continuously to determine the grain number of the panicle. The 1,000-grain weight was air-dried, weighed, and adjusted to a standard 12% moisture content. All the above measurements were repeated three times per treatment.

The corresponding date of the main growth period of wheat was recorded in the field according to Zadok's growth scale (*Karimizadeh & Mohammadi, 2011*) with more than 50% of seedlings in the plot sampled. We collected the daily maximum, minimum, and average temperature, light hours, daily rainfall, and other meteorological datum during the two-year wheat growth period from the Jingzhou Meteorological Bureau of Hubei Province.

## Calculations and data analysis

A logistic model was used to quantitatively describe the dynamic changes of accumulation in winter wheat shoots to model the accumulation pattern as follows (*Du et al., 2016*):

$$W = \frac{W_{max}}{1 + ae^{-kT}} \tag{1}$$

where, $W$ (g stem$^{-1}$) is the accumulation in wheat, $W_{max}$ (g stem$^{-1}$) is the theoretical maximum accumulation, $T$ (d) is days after emergence, a and k are the constants to be found.

The following functions can be obtained by calculating the first, second, and third derivative of Eq. (1), respectively.

$$T_1 = \frac{1}{k} ln \frac{2 + \sqrt{3}}{a} \tag{2}$$

$$T_2 = \frac{1}{k} ln \frac{2 - \sqrt{3}}{a} \tag{3}$$

$$T_m = \frac{lna}{k} \tag{4}$$

$$V_m = \frac{kW_{max}}{4} \tag{5}$$

where, $T_1$ (d) is the accumulated growth time at the fastest beginning date of growth curve, $T_2$ (d) is the accumulated growth time at the termination date, maximum relative growth rate $V_m$ (g stem$^{-1}$ d$^{-1}$) and its cumulative growth time $T_m$ (d).

The fast accumulation phase began at $T_1$ and ended at $T_2$, $W$ is a linear correlation with the days after emergence and the average growth rate ($V_t$).

$$V_t = \frac{W_2 - W_1}{T_2 - T_1}. \tag{6}$$

Yield loss (%) due to the early or late sowing was calculated as follows (*Shah et al., 2020*):

$$Yield\ loss(\%) = \frac{Y_{ns} - Y}{Y_{ns}} \times 100 \tag{7}$$

where $Y_{ns}$ and $Y$ are the grain yields of normal and early or late sowing dates, respectively.

## Statistical analysis

Data preparation was performed with Microsoft Excel 2010 and the final data plots were produced with Origin 8.0. Multiple comparisons were performed after a preliminary F-test. Means were tested based on the least significant difference at $P < 0.05$ using Data Processing System (DPS) v.7.05 software.

# RESULTS

## Weather conditions and crop phenology

The mean daily temperatures of the two wheat growing seasons first decreased and then increased (Fig. 1). The mean daily temperatures below 10 °C lasted 89 days (for a continuous three days or longer, lower than 10 °C) and decreased to 77 days in the second year. The thermal time from the sowing to wintering stage decreased greatly when sowing was delayed by more than eight days across two experimental years (Table 2). The average thermal time was reduced by 21%, 37%, and 51% for 8–, 16–, and 24–days delay in sowing compared with the first sowing date (28-Oct).
**Table 2   Statistics of growth time and meteorological data of different sowing dates in growing season of 2018–2019 and 2019–2020.**

| Season | Sowing date | Days for sowing to flowering stage (d) | Days for filling stage (d) | Accumulated temperature during the whole growth period (°C) | Thermal time from sowing to wintering stage (°C d) | Mean daily temperature during filling stage (°C) |
|---|---|---|---|---|---|---|
| 2018–2019 | 28-Oct | 164 | 38 | 2,141.2 | 623.0 | 20.08 |
|  | 5-Nov | 159 | 37 | 2,048.6 | 486.2 | 20.28 |
|  | 13-Nov | 153 | 36 | 1,978.9 | 397.6 | 20.48 |
|  | 21-Nov | 149 | 36 | 1,985.2 | 310.2 | 20.71 |
| 2019–2020 | 28-Oct | 146 | 50 | 2,304.6 | 741.6 | 17.93 |
|  | 5-Nov | 143 | 46 | 2,187.1 | 601.6 | 18.23 |
|  | 13-Nov | 142 | 43 | 2,140.7 | 469.3 | 19.84 |
|  | 21-Nov | 138 | 39 | 2,028.8 | 357.4 | 20.25 |

The mean daily temperatures increased gradually from flowering to the end of grain-filling when sowing was delayed by more than 8 days across two experimental years. The temperatures ranged from 20.08 °C for 28-Oct to 20.71 °C for 21-Nov in 2018–2019, and 17.93 °C for 28-Oct to 20.25 °C for 21-Nov in 2019–2020 (Table 2). There was a significant negative correlation in the second year ($r = -0.94$, $P < 0.05$) between the grain filling days of each sowing date and the mean daily temperatures during the filling stage.

The crop growth cycle from seeding to the end of the grain filling period of each sowing date was significantly shortened when the sowing date was delayed in the 2018–2019 and 2019–2020 growing seasons. The growth duration of 8–, 16–and 24–days delay in sowing decreased by 7, 12, and 18 days on average over two growing seasons when compared with the first sowing date (28-Oct). This difference was mainly due to the earlier flowering period and the compression of the grain-filling period for the late sowing date. The average flowering date moved forward by 4.0, 7.5, and 11.5 days for an 8–, 16–, and 24–day delay in sowing compared with the first sowing date (28-Oct). The duration of the filling period of the four sowing dates in the first year was not significantly shortened. In the second year the latest sowing (21-Nov) and earliest sowing (28-Oct) shortened the filling period by 11 days under a delay of 24 days (Table 2).

## Morphological traits

There were differences in the number of tillers at jointing and maturity. The tiller number at the jointing stage of the latest sowing date (21-Nov) was higher than other sowing dates over the two growing seasons. The tiller number of the 21-Nov sowing date was significantly different from that of other sowing dates during the second year. The peak tillers appeared before or after the jointing stage. The peak tillers appeared before the jointing stage for the early sowing date (28-Oct) and the normal sowing date (5-Nov), while the late sowing date (13- Nov) and the latest sowing date (21-Nov) occurred after the jointing stage, which was consistent for both years (Table 3). The results of our two-year experiments showed that the tiller number of the 5-Nov sowing date was significantly higher than that of other sowing dates at maturity stage. The percentage of productive tillers on 13-Nov was significantly higher than the other sowing dates. Delayed sowing caused the percentage of productive

**Table 3** The number of tillers and effective tiller rate of winter wheat at different sowing dates in 2018–2019 and 2019–2020 growing seasons.

| Season | Sowing date | Tiller number at jointing stage (no. m$^{-2}$) | Tiller number at maturity stage (no. m$^{-2}$) | Peak tillers (no. m$^{-2}$) | Percentage of productive tillers (%) |
|---|---|---|---|---|---|
| 2018–2019 | 28-Oct | 530.67b | 444.00b | 785.33a | 56.65c |
| | 5-Nov | 636.00a | 484.33a | 754.67a | 64.23b |
| | 13-Nov | 596.00ab | 437.67b | 596.00b | 73.45a |
| | 21-Nov | 640.00a | 371.67c | 640.00b | 58.33bc |
| 2019–2020 | 28-Oct | 524.00c | 432.00b | 1,078.67a | 40.06c |
| | 5-Nov | 642.33b | 466.67a | 876.00b | 53.48b |
| | 13-Nov | 660.00b | 411.67c | 660.00c | 62.38a |
| | 21-Nov | 701.33a | 391.33d | 701.33c | 55.80b |

**Notes.**
Values followed by the same letter within a column in the same year are not significantly different at $P < 0.05$ as determined by the LSD test.

tillers to increase significantly each 8-day delay in sowing except for the latest sowing date (21-Nov).

## Yield formation

The wheat yield varied by both year and sowing date and these factors interacted significantly (Table 4). The grain yield for different sowing dates ranged from 5,569.7 to 6,578.9 kg ha$^{-1}$ in 2018–2019 and from 5,625.0 to 7,241.7 kg ha$^{-1}$ in 2019–2020. Grain yield for each sowing date in 2019–2020 was 1.0%~10.1% greater than that in 2018–2019. Grain yield from the 5-Nov sowing date was greater than those of the other treatments during both years. The average grain yield from 5-Nov was 3.2%, 18.7% and 23.4% greater than the yields of 28-Oct, 13-Nov, and 21-Nov, respectively. The results of two-year experiments showed that the yield decreased significantly when the sowing date was delayed for 8 and 16 days after 5-Nov. A consistent declining trend was observed for grain yield after this date until the last sowing date during both years, showing that the longer the delay in sowing date, the greater the yield reduction (Table 4 and Fig. 2). Regression analysis revealed that with each day that grain was sowed early or delayed, the grain yield declined by 0.97 ± 0.22% across 2 years (Fig. 2).

The spike number was affected ($P < 0.01$) by year and sowing date, and the interaction was extremely significant (Table 4). Kernel number per spike reached a very significant level ($P < 0.01$) by year and sowing date, but the interaction term was not significant. 1,000-kernel weight was only affected ($P < 0.01$) by year, and there was no significant difference between the sowing date and these factors. The spike number was positively correlated with wheat grain yield ($r = 0.73$, $P < 0.05$), whereas the kernel number and 1,000-kernel weight were not. The spike number per ha for 5-Nov was 9.1%~30.3% in 2018–2019 and 8.0%~19.3% in 2019–2020, which was greater than that for 28-Oct, 13-Nov and 21-Nov, respectively. The kernel per spike for 21-Nov was 0.5%, 8.5% and 8.2% in 2018–2019, and 2.7%, 8.0% and 3.7% in 2019–2020, which was greater than that for 28-Oct, 5-Nov and 13-Nov, respectively.

**Table 4  Grain yield and yield components for different sowing dates in 2018–2019 and 2019–2020 growing seasons.**

| Season | Sowing date | Grain yield (kg ha⁻¹) | Spike number (10⁴ ha⁻¹) | Kernel number per spike | 1000-kernel weight (g) |
|---|---|---|---|---|---|
| 2018–2019 | 28-Oct | 6,403.8a | 444.0b | 40.8a | 40.5a |
| | 5-Nov | 6,578.9a | 484.3a | 37.8b | 40.6a |
| | 13-Nov | 5,674.8b | 437.7b | 37.9b | 40.9a |
| | 21-Nov | 5,569.7b | 371.7c | 41.0a | 41.1a |
| 2019–2020 | 28-Oct | 6,980.0b | 432.0b | 40.9b | 46.9a |
| | 5-Nov | 7,241.7a | 466.7a | 38.9c | 47.2a |
| | 13-Nov | 5,958.3c | 411.7c | 40.5b | 47.7a |
| | 21-Nov | 5,625.0d | 391.3d | 42.0a | 48.1a |
| Year | | .0001 | .0002 | .0010 | .0001 |
| SD | | .0001 | .0001 | .0001 | .1425 |
| Y ×SD | | .0003 | .0001 | .0627 | .8184 |

Notes.

Values followed by the same letter within a column in the same year are not significantly different at $P < 0.05$ as determined by the LSD test.

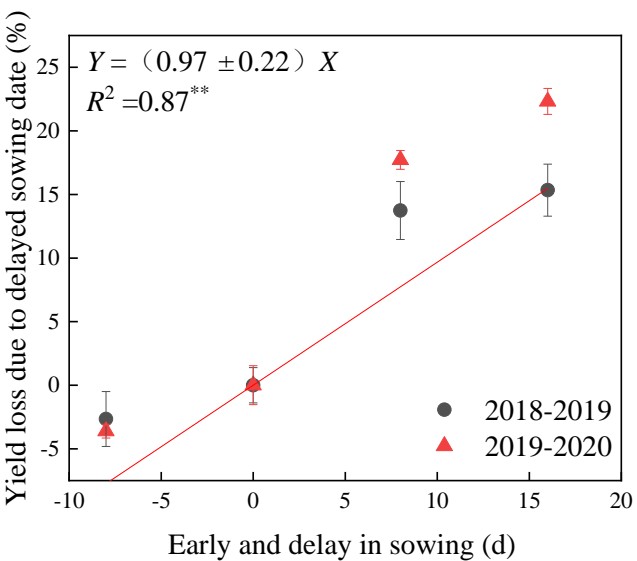

**Figure 2  Grain yield loss of winter wheat due to the early and delayed sowing during the 2018–2019 and 2019–2020 growing seasons.** Vertical bars indicate standard errors ($n = 3$).

## Dynamics simulation of DMA

We showed that the dynamic changes of DMA for single stem plants with the days after sowing conforms to the logistic curve model (Fig. 3). The logistic function was followed by DMA as a sigmoidal growth pattern since all $P$ values were < 0.01 (Table 5), although they differed in equation coefficients among the treatments. The simulated value of DMA for single stem plants was determined using formula (1), and we were able to obtain the characteristic value of dynamic of DMA for single stem plants (Table 6).

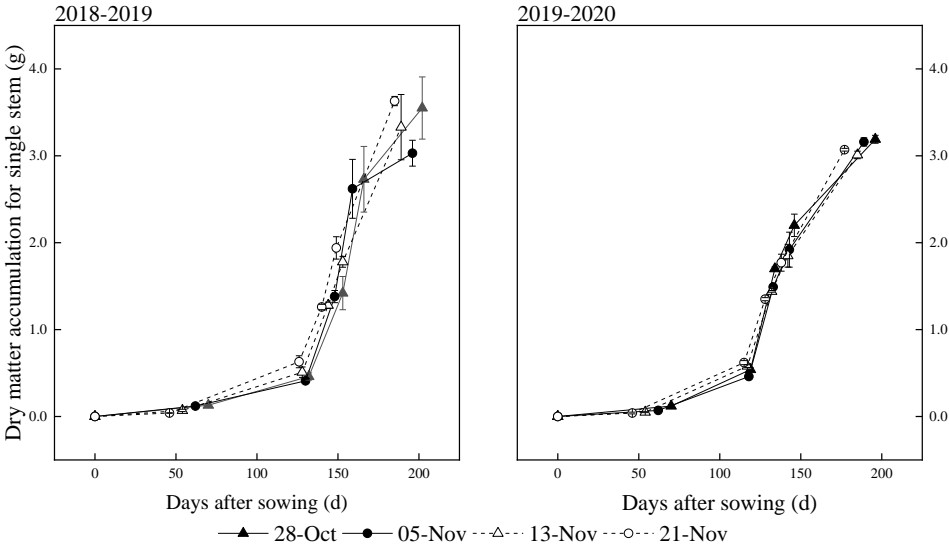

**Figure 3** **Effects of different sowing dates on DMA for a single stem in 2018–2019 and 2019–2020.** Values are means of three replicates per treatment. Vertical bars indicate standard error.

**Table 5** **Equations of DMA under different sowing dates in 2018–2019 and 2019–2020 growing seasons.**

| Season | Sowing date | Regression equations | $R^2$ |
|---|---|---|---|
| 2018–2019 | 28-Oct | $Y = 3.6217/(1+4101240.31e^{-0.0976t})$ | 0.9941[**] |
| | 5-Nov | $Y = 3.0897/(1+420518907.20e^{-0.1339t})$ | 0.9909[**] |
| | 13-Nov | $Y = 3.5946/(1+45779.88e^{-0.0701t})$ | 0.9992[**] |
| | 21-Nov | $Y = 3.9149/(1+48761.68e^{-0.0722t})$ | 0.9995[**] |
| 2019–2020 | 28-Oct | $Y = 3.1812/(1+109524.11e^{-0.0862t})$ | 0.9925[**] |
| | 5-Nov | $Y = 3.1866/(1+91308.58e^{-0.0836t})$ | 0.9949[**] |
| | 13-Nov | $Y = 3.0814/(1+20681.61e^{-0.0734t})$ | 0.9977[**] |
| | 21-Nov | $Y = 3.2251/(1+10799.99e^{-0.0692t})$ | 0.9981[**] |

**Notes.**
[**] Significant differences at $P < 0.01$ probability levels ($n = 6$).

We found the beginning and termination day at the rapid accumulation period of winter wheat DMA for single stem under all sowing dates during both years using formulas (2)–(4) based on Table 5. The start of the rapid accumulation period was noted as being after the jointing stage and the termination day was considered to be after the flowering stage. The value of the fastest accumulation period over two years was shorter in early sowing (27.0 ∼30.6 d) and normal sowing (19.7 ∼31.5 d) than in late sowing (35.9 ∼37.5 d) and the latest sowing (36.5 ∼38.1 d). There were differences among the treatments of DMA for single stem plants under different sowing dates in both years. The maximum relative growth rates ($V_m$) and highest average rates ($V_t$) of early sowing and normal sowing were slightly higher than those of late sowing and the latest sowing. The fastest DMA point of single stem growth was during the booting and flowering stages.

**Table 6  Effects of different sowing dates on the eigenvalues of DMA in the 2018–2019 and 2019–2020 growing seasons.**

| Season | Sowing date | Fast accumulation period | | | | Fastest accumulation point | |
|---|---|---|---|---|---|---|---|
| | | $T_1$ (d) | $T_2$ (d) | $T$ (d) | $V_t$ (g stem$^{-1}$ d$^{-1}$) | $T_m$ (d) | $V_m$ (g stem$^{-1}$ d$^{-1}$) |
| 2018–2019 | 28-Oct | 142.5 | 169.5 | 27.0 | 0.08 | 156.0 | 0.09 |
| | 5-Nov | 138.5 | 158.2 | 19.7 | 0.09 | 148.4 | 0.10 |
| | 13-Nov | 134.2 | 171.8 | 37.5 | 0.06 | 153.0 | 0.06 |
| | 21-Nov | 131.4 | 167.9 | 36.5 | 0.06 | 149.6 | 0.07 |
| 2019–2020 | 28-Oct | 119.4 | 150.0 | 30.6 | 0.06 | 134.7 | 0.07 |
| | 5-Nov | 120.9 | 152.4 | 31.5 | 0.06 | 136.6 | 0.07 |
| | 13-Nov | 117.4 | 153.3 | 35.9 | 0.05 | 135.4 | 0.06 |
| | 21-Nov | 115.3 | 153.3 | 38.1 | 0.05 | 134.3 | 0.06 |

**Notes.**
$T_1$ and $T_2$, Beginning and termination days of the duration of fast accumulation phase; $T$ $T_2$- $T_1$, Duration of the physiological development time in rapid accumulation period; $V_t$, Average accumulation rate during the duration of fast accumulation phase; $T_m$, Accumulation of physiological development time reached maximal accumulation rate; $V_m$, Maximum accumulation rate during the duration of fast accumulation phase, respectively.

Our results showed that $V_m$ and $V_t$ of early sowing and normal sowing were higher than that of late sowing and the latest sowing. The dynamic accumulation characteristic parameters of DMA for single stem were better and produced better biomass accumulation and yield formation.

## Dynamics simulation of NA

The dynamic changes of NA for single stem plants after sowing were consistent with the changes of DMA. The dynamic changes of NA for single stem conformed to the logistic curve model as the plants grew. It can be seen from Fig. 4 that delayed sowing promotes the absorption of N for single stems of winter wheat, and NA for single stems also increased with the delay of sowing date.

The beginning and termination of the rapid period of accumulation of NA for single stems occurred before the jointing stage and after the booting stage, respectively, according to formulas (2)–(4) based on Table 7. The fastest NA point was from the jointing to the booting stage for different sowing dates in both years. The duration of fast NA for single stem was gradually shortened with the delay of the sowing date. Normal sowing (5-Nov) was postponed by 8 and 16 days, and the average accumulation period was shortened by 0.8 days and 5.2 days. The later the sowing date, the shorter the period of rapid accumulation (Table 8). The maximum relative growth rates ($V_m$) and highest average rates ($V_t$) of NA for single stems were different over two years, which showed an increasing trend with the delay of sowing date. The maximum relative growth rates ($V_m$) and highest average rates ($V_t$) reached the peak at the latest sowing.

Our results showed that with the delay of sowing date, the coordination of dynamic accumulation characteristic parameters for single stem NA of late sowing wheat was better than that of other treatments. The start of the rapid accumulation period was earlier than that of DMA, indicating that the biomass grew based on adequate nutrient absorption.

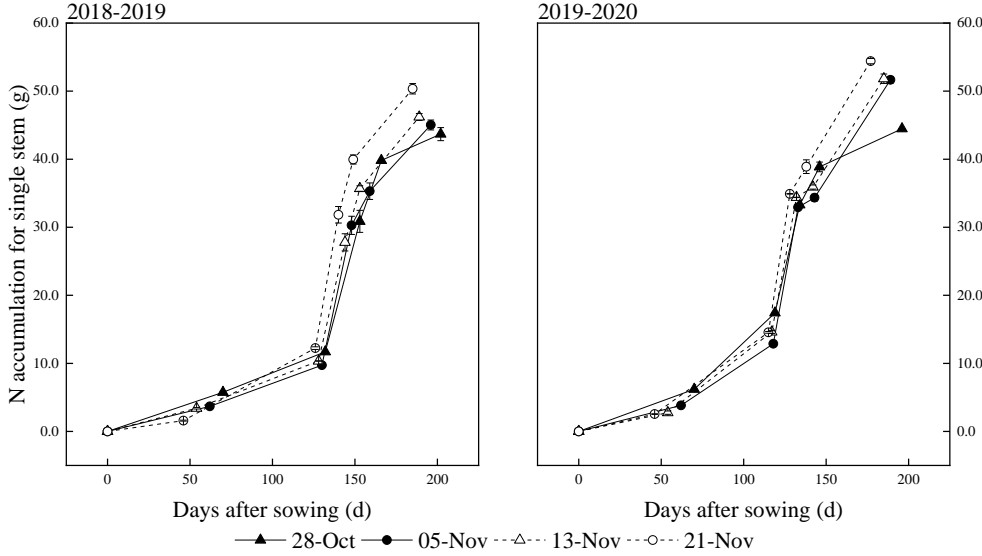

**Figure 4 Effects of different sowing dates on NA dynamics for a single stem in 2018–2019 and 2019–2020.** Values are means of three replicates per treatment. Vertical bars indicate standard error.

**Table 7 Equations of NA under different sowing dates in 2018–2019 and 2019–2020 growing seasons.**

| Season | Sowing date | Regression equations | $R^2$ |
|---|---|---|---|
| 2018–2019 | 28-Oct | $Y = 44.2207/(1+436306.39e^{-0.0906t})$ | 0.9808[**] |
| | 5-Nov | $Y = 44.6148(1+839280.29e^{-0.0960t})$ | 0.9874[**] |
| | 13-Nov | $Y = 46.4937/(1+1030959.01e^{-0.0986t})$ | 0.9932[*] |
| | 21-Nov | $Y = 50.2932/(1+3464573.17e^{-0.1109t})$ | 0.9980[*] |
| 2019–2020 | 28-Oct | $Y = 44.8498/(1+28575.36e^{-0.0834t})$ | 0.9787[**] |
| | 5-Nov | $Y = 51.9671/(1+12335.05e^{-0.0724t})$ | 0.9782[**] |
| | 13-Nov | $Y = 52.0154/(1+12551.56e^{-0.0741t})$ | 0.9838[**] |
| | 21-Nov | $Y = 54.2685/(1+62975.43e^{-0.0888t})$ | 0.9872[**] |

**Notes.**
[*]Significant differences at $P < 0.05$ probability levels ($n = 6$).
[**]Significant differences at $P < 0.01$ probability levels ($n = 6$).

## Relationships among grain yield, $M_D/J_D$, and $M_N/J_N$

Since 5-Nov, the aboveground biomass and N production were significantly reduced with an increase in the number of days that sowing was delayed (Fig. 5). The average DMA decreased by 8.50% and 13.32% after 8 and 16 day delays from normal sowing. The rule of NA remained the same, decreasing by 9.40% and 12.95% after 8 and 16 days delayed of normal sowing.

We performed correlation analyses to identify the grain yield associated with the law of DMA and NA. The ratio of DMA at the mature to jointing stages was recorded as $M_D/J_D$ and the ratio of NA at the mature to jointing stages was recorded as $M_N/J_N$. Correlations were observed between the grain yield and $M_D/J_D$, and both years showed a very significant positive correlation (Figs. 6A and 6B). Furthermore, there was a significant

**Table 8  Effects of different sowing dates on the eigenvalues of NA in the 2018–2019 and 2019–2020 growing seasons.**

| Season | Sowing date | Fast accumulation period | | | | Fastest accumulation point | |
|---|---|---|---|---|---|---|---|
| | | $T_1$ (d) | $T_2$ (d) | $T$ (d) | $V_t$ (g stem$^{-1}$ d$^{-1}$) | $T_m$ (d) | $V_m$ (g stem$^{-1}$ d$^{-1}$) |
| 2018–2019 | 28-Oct | 128.8 | 157.9 | 29.1 | 0.88 | 143.3 | 1.00 |
| | 5-Nov | 128.4 | 155.8 | 27.4 | 0.94 | 142.1 | 1.07 |
| | 13-Nov | 127.0 | 153.7 | 26.7 | 1.01 | 140.4 | 1.15 |
| | 21-Nov | 123.9 | 147.7 | 23.8 | 1.22 | 135.8 | 1.39 |
| 2019–2020 | 28-Oct | 107.3 | 138.8 | 31.6 | 0.82 | 123.1 | 0.93 |
| | 5-Nov | 111.9 | 148.3 | 36.4 | 0.82 | 130.1 | 0.94 |
| | 13-Nov | 109.6 | 145.1 | 35.5 | 0.85 | 127.3 | 0.96 |
| | 21-Nov | 109.6 | 139.3 | 29.7 | 1.06 | 124.5 | 1.20 |

**Notes.**

$T_1$ and $T_2$, Beginning and termination days of the duration of fast accumulation phase; $T$ $T_2$- $T_1$, Duration of the physiological development time in rapid accumulation period; $V_t$, Average accumulation rate during the duration of fast accumulation phase; $T_m$, Accumulation of physiological development time reached maximal accumulation rate; $V_m$, Maximum accumulation rate during the duration of fast accumulation phase, respectively.

positive correlation between yield and $M_N/J_N$ (Figs. 6C and 6D). This suggests that the two ratios have an important influence on the grain yield of winter wheat under different sowing dates.

The linear relationship was found between the grain yield and $M_D/J_D$ ($Y = 306.03X$ $+4511.69$ in 2018–2019 and $Y = 636.93X$ $+3924.05$ in 2019–2020) and $M_N/J_N$ ($Y = 698.98X$ $+3904.74$ in 2018–2019 and $Y = 1273.29X$ $+3485.91$ in 2019–2020) (Fig. 6). As $M_D/J_D$ and $M_N/J_N$ increased, the grain yield increased. The grain yield could be maintained at 6,000 kg ha$^{-1}$ or above when the average DMA and NA reached 4.06 ($P < 0.01$) and 2.49 ($P < 0.05$), respectively.

## DISCUSSION

### Effects of different sowing dates on yield

We present new data to support the common perception that the sowing date is a crucial agronomic factor for improving, growth, grain yield, and nutrient acquisition of winter wheat. Advancing or delaying beyond the optimum sowing time may hinder the full genetic yield potential of winter wheat. This study evaluated grain yield changes and the biological effects caused by different sowing dates. The grain yield declined by 0.97 ± 0.22% with each day that sowing was conducted early or late beyond the normal sowing date (Fig. 2). Previous studies have determined the grain yield declined as a result of delayed sowing with an average yield penalty of approximately 0.37 ± 0.07% with each one-day delay in sowing beyond the normal sowing date (*Yin, Dai & He, 2018*; *Ma et al., 2018*; *Dwivedi et al., 2019*; *Dubey et al., 2019*; *Zhu et al., 2019*; *Gandjaeva, 2019*). Those results are comparable with ours. Grain yield reduction as a result of delayed sowing can be explained in terms of suppressed of crop growth, and decreased spike number, dry matter, and N production.

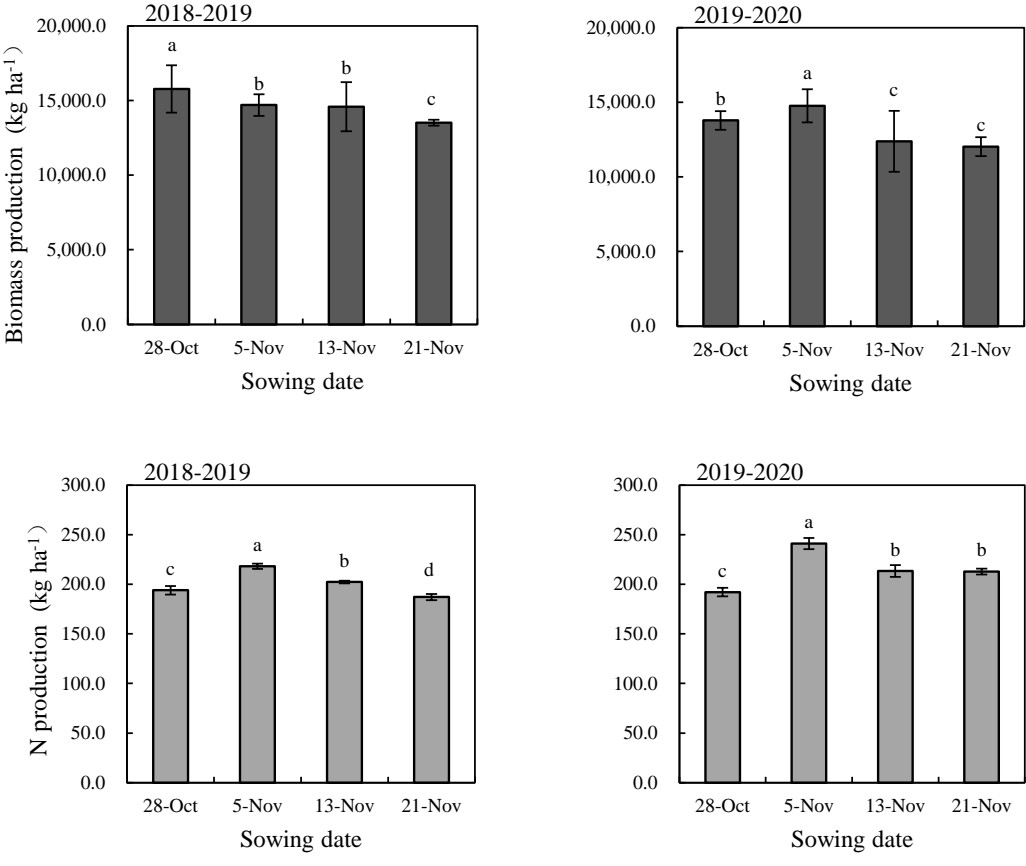

**Figure 5** **Biomass production and N production of winter wheat under different sowing date during the 2018–2019 and 2019–2020 growing seasons.** Vertical bars indicate standard errors ($n = 3$). Means followed by the same letter are not significantly different among sowing dates according to the LSD_0.05 test.

## Changes in weather conditions and crop phenology on different sowing dates

Environmental factors significantly affected the grain yield of wheat that was sown late. It is clear that under late sowing conditions, plants faced adverse weather factors such as low temperature and less thermal exposure during sowing to wintering stage compared with the normal sowing date (Fig. 1). For example, the thermal time from the sowing to the wintering period decreased an average of 543.9 °C d in normal sowing to 333.8 °C d after 16-days of delayed sowing (Table 1). These disadvantages may affect early crop growth by inhibiting seed emergence, seedling establishment, and tiller development (*Shah et al., 2019*; *Zhou et al., 2020*). A delay in sowing will also lead to earlier flowering, reducing the duration of various phases of crop development. The results showed that under the condition of the latest sowing, the crop growth duration from sowing to flowering was shortened by 7.5 days, compared to the normal sowing (Table 2). Shortening of the critical phenological period, as a key factor in determining the photoperiod and productivity of crops, may further explain the poor performance under delayed sowing (*Ferrise et al., 2010*; *Sattar et al., 2010*). Delayed sowing also increases the possibility of crops exposure to high

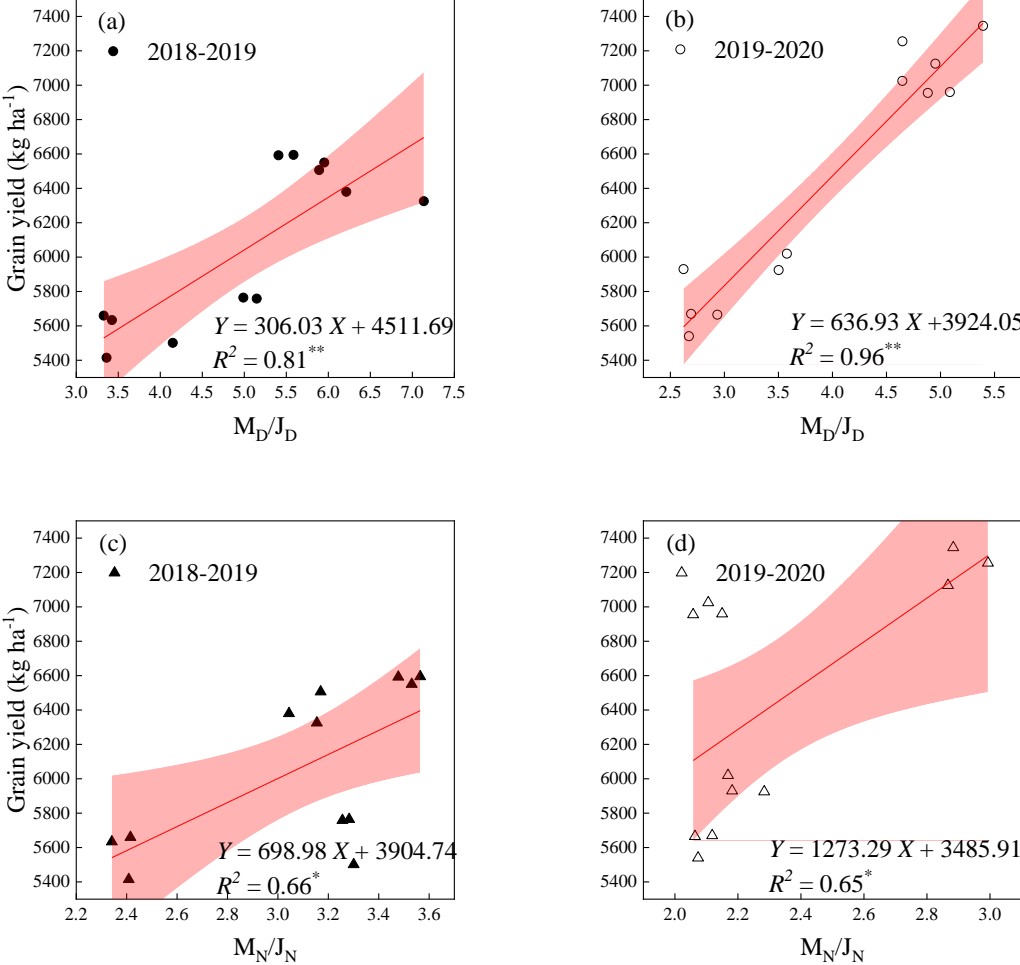

**Figure 6 Relationships between grain yield and four ratios.** (A) the ratio of DMA at maturity stage to jointing stage ($M_D/J_D$) in 2018–2019; (B) the ratio of DMA at maturity stage to jointing stage ($M_D/J_D$) in 2019-2020; (C) the ratio of NA at maturity stage to jointing stage ($M_N/J_N$) in 2018–2019; (D) the ratio of NA at maturity stage to jointing stage ($M_N/J_N$). * correlation coefficients are significant at the 0.05 probability level; ** correlation coefficients are significant at the 0.01 probability level. The shaded part represents the 95% confidence band.

temperature during the grain filling stage, which is harmful to leaf photosynthesis, grain filling, and final yield formation. These are considered to be the key stressors for wheat production in many environments around the world (*Garg et al., 2013*).

## Different sowing dates affects tillers

The environmental limitations posed by late sowing, the inhibition of early growth, and a shorter vegetative growth period led to low tillering ability, poor tiller development, and the reduction of productive tillers. Although late sowing could establish a greater tiller population around the jointing stage, it did not maintain this advantage throughout the growth period. Compared with normal sowing, the tillers decreased significantly at the mature stage, which reduced the percentage of productive tillers (Table 3). The increased
seeding rate could make up for the decrease of tillers in the late-sown winter wheat (*Wang et al., 2016*; *Ma et al., 2018*). In our study, wheat was planted at a constant density to eliminate the effects of density on wheat growth and grain yield, resulting in the change of spike number with the delay in sowing because of the decreased tillers (*Xu et al., 2018*; *Zhu et al., 2019*). However, the reduced plant population could increase the number of fertile stems per plant and the number of kernels per spike, but the magnitude is less than the grain weight (*Whaley et al., 2015*). This is consistent with our results that the spike number and kernel number were the main factors influencing grain yield. However, there is no correlation between grain yield and 1,000-kernel weight (Table 4).

## Different sowing dates affects DMA and NA

DMA and NA are two primary factors influencing wheat grain yield and grain quality. The demand for high-yielding and high-quality wheat is expected to increase dramatically in the near future (*Meng et al., 2013*; *Jin et al., 2018*). Reducing the vegetative growth period and tiller number could explain the significant decrease in DMA and final yield under delayed sowing (*Shah et al., 2020*). The number of tillers decreased significantly with a delay in the sowing date. DMA and NA for single stems played an important role in grain yield and quality. The longest duration of the delayed sowing was longer than that of normal sowing date at the rapid accumulation period of DMA for single stems. This serves to maintain the corresponding accumulation amount although the maximum relative growth rates ($V_m$) and highest average rates ($V_t$), which showed that the normal sowing date was slightly higher than the delayed sowing. However, NA for single stems showed the opposite effect. With the delay of sowing date, $V_m$ and $V_t$ increased gradually, and peaked at the latest sowing. The compensation effect of the rapid accumulation period also showed that the accumulation period was compressed when the sowing date was delayed. There was a relationship between the growth rate of dry matter and the amount of nutrient absorption, but they were not synchronous. The maximum rate of NA occurs earlier than that of DMA (*Song & Li, 2003*). In this study, the accumulation of physiological development time reached maximal accumulation rate ($T_m$) of DMA for winter wheat 6.4~12.2 days later than that of NA (Tables 6 and 8). Therefore, it was necessary to apply N fertilizer at the jointing stage to ensure the nutrient absorption of wheat dry matter and N during the rapid accumulation period. Our results showed that the final biomass and N production of the treatment with normal sowing date were significantly higher than those with delayed sowing (Fig. 5), which was consistent with the results of previous studies (*Yin, Dai & He, 2018*; *Ferrise et al., 2010*).

## Directions for future study

We reiterate the need for sowing at the most suitable time for the maturity length and growing season for a particular variety, which is considered critical for yield optimization. Our results show that sowing in the optimum range can result in a higher yield and that as the sowing date is delayed, the grain yield continues to decline. The highest grain yield was produced with seeds sown on 5-Nov, which occurred due to better weather conditions and a reasonable distribution of thermal time during the wheat growing seasons over two years.

The sowing date is an agronomic measure regulated by genotypes and is heavily dependent on the location and environment of crop growth. It is imperative to conduct further multi-location experiments with a diverse set of winter wheat genotypes. A significant aspect would be to compare the yield compensation of different winter wheat genotypes with different tillering capacities and crop growth durations, because these were the most critical traits influenced in response to delayed sowing. Future studies should focus on increasing the seeding rate to compensate for the winter wheat yield reduction caused by delayed sowing.

## CONCLUSION

Grain yield declined by $0.97 \pm 0.22\%$ as grains were sown with each one-day early or delay in sowing beyond the normal sowing date. This yield penalty may be due to the inhibition of crop growth, yield components, biomass, and N production. The negative effects of delayed sowing are mainly caused by key environmental limitations including adverse weather factors such as low temperature during vegetative growth, shortened duration of various phases of crop development and increased temperature during grain filling period. In other words, the sowing date determines the weather conditions to which wheat is exposed. However, owing to a compensation effect between the highest average rates ($V_t$) and the rapid accumulation period ($T$) of DMA and NA for single stems, the grain yield gap decreased between late and normal sowing. Under these conditions, if the ratio of $M_D/J_D$ and $M_N/J_N$ could reach 4.06 ($P < 0.01$) and 2.49 ($P < 0.05$), respectively, grain yield could be maintained at the level of 6,000 kg ha$^{-1}$ or above. However, delayed sowing caused biomass and N production to decline. Meanwhile, the accumulation of physiological development time reached maximal accumulation rate ($T_m$) of NA earlier than that of DMA. We conclude that the reasonable sowing date of winter wheat in the middle-lower Yangtze River Basin around 5-Nov. Further research is needed to explore the compensatory effect of different genotypes of winter wheat on yield under the condition of increasing sowing rate if late sowing is inevitable.

### Funding

This research was supported by the National Natural Science Foundation of China (31871578) and the National Key Research and Development Program of China (2016YFD0300107, 2017YFD0300205). There was no additional external funding received for this study. The funders had no role in study design, data collection and analysis, decision to publish, or preparation of the manuscript.

### Grant Disclosures

The following grant information was disclosed by the authors:
National Natural Science Foundation of China: 31871578.
National Key Research and Development Program of China: 2016YFD0300107, 2017YFD0300205.

## Competing Interests

The authors declare there are no competing interests.

## Author Contributions

- Kaizhen Liu, Lijun Yin and Xiaoyan Wang conceived and designed the experiments, performed the experiments, analyzed the data, prepared figures and/or tables, authored or reviewed drafts of the paper, and approved the final draft.
- Chengxiang Zhang, Beibei Guan, Rui Yang, Ke Liu, Zhuangzhi Wang, Xiu Li and Keyin Xue performed the experiments, prepared figures and/or tables, and approved the final draft.

## Data Availability

Raw data are available in the Supplemental File.

## Supplemental Information

Supplemental information for this article can be found online at http://dx.doi.org/10.7717/peerj.11700#supplemental-information.

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
