# Peer review of "The effect of different sowing dates on dry matter and nitrogen dynamics for winter wheat: an experimental simulation study"

_PeerJ, doi:10.7717/peerj.11700_

## Round 0.1 · original submission · Major Revisions

· Academic Editor

Major Revisions

Dear authors, to improve the quality of your manuscript, I suggest making the improvements indicated by the reviewers, which are mostly in form, although in this second round it will be substantially improved.

Reviewer 1 ·

Basic reporting

Fig.1 , X axis indication are missed, please report date in term of month, if the months are considered. /
L166-171. You describe the plant samples and the N determination. It is clear that you sampling the different plant part, leaves, stem etc, what is not clear and not specified if you determine the N on the single parte , i.e. leaves, or if you , after dry mix the different part to get a single samples. The procedure is not clear, even if later you report that the N accumulation is the sum of the different organ. Please rewrite this paragraph .
Check caption table 5-7

In several part there are spelling error, e.g eigen value =eigenvalue

Experimental design

no comment

Validity of the findings

no comment

Additional comments

General comment
The manuscript is interesting but some section are not clear in English. Therefore, I suggest a English revision and an improvement in clarity in material and methods. This minor revision will make the manuscript better in quality and able to be publish.

Annotated reviews are not available for download in order to protect the identity of reviewers who chose to remain anonymous.

·

Basic reporting

1. Manuscript is written in standard English.
2. Literature References are insufficient and some of them are too old (references from 1959 is mentioned).
3. Article structures need to be improved, figure resolution is to be improved.
4. Result and Discussion section is also need to be improved.

Experimental design

Does not meet entirely.... Needs improvement (specifically shared with Author's section).

Validity of the findings

Result and Discussion need to be improved (specifically shared with Author's section).

Additional comments

The research article entitled Different sowing dates effect on dry matter and nitrogen dynamics for winter wheat: An experimental study based on simulation is a quite well documented approach by Liu et. al., in terms of assessing the effects of different sowing dates on different agronomic parameters of winter wheat and checking relevance of a logistics model in quantifying Dry Matter accumulation (DMA) and Nitrogen Accumulation (NA) in winter wheat. As the manuscript was observed further, some noticeable rearrangement issues of this article are needed to be rectified. These are as follows-
1. In the Abstract section, please mention the novelty and objectives of this research work in the ‘Background’ part and mention the variables of the experiments for each objectives within the ‘Methods’ section. Also mention the novelty in clear cut sentence within the “Introduction” portion as well.
2. Please provide the full form of all of the abbreviations during their first mention, and then apply abbreviations within the text. Please also write ‘Nitrogen Accumulation’ in place of ‘N Accumulation’.
3. Please provide the initial physicochemical characteristics of the soil in a separate table.
4. Please provide the bibliographic reference to the fertilizer application practice followed. Please also explain in brief why an additional 90 kg ha-1 of N was applied at the beginning of the jointing growing stage?
5. Please provide weather and climatic information (max-min temperatures, RH and rainfall pattern and daily sunshine hour) with proper bibliographic refetence, if not recorded by researcher.
6. Please also prove the fertilizer application history/pattern of that area (include previous crops also).
7. Please mention ‘Experimental Design’ properly… whether its CRD/RBD or Split plot… Please mention that.
8. Please also provide suitable bibliographic reference to the analytical procedures followed in Measurement items and methods.
9. Please mention the version of M.S. Excel.
10. Please mention and provide tables for the information mentioned under ‘Weather conditions and crop phenology’ and ‘Morphological traits’.
11. Please increase the resolution of the figures and make them bigger in size so that each detail is easily observable.
12. It is highly appreciable if the authors could improve the discussion section. The entire Discussions section is highly inconsistent in relating the results with the objectives to be fulfilled. Please take care of the following points within the Discussion segment-
a. Please address the findings of the Result section in relation with the fulfillment of the objectives.
b. Please add more relevant references of previous works and relate the current research.
c. At the end of the discussion, mention in clear cut sentences, how far the objectives are fulfilled and also mention of there is any future scope of research.
d. Write the conclusion mentioning the main points from the reformed Discussion segment.
13. It is really appreciable to observe the low plagiarism of the manuscript (9%) as detected by Plagiarism Checker X (report attached).

---

## Round 0.2 · accepted · Accept

· Academic Editor

Accept

I have considered that the comments of the reviewers and the improvements that the authors have added have been incorporated to improve the quality of their manuscript. My sincere congratulations for the efforts of the authors to make the improvements and the professional scientific aspect that they had to do so.